# Ibrutinib Associated with Rituximab-Platinum Salt-Based Immunochemotherapy in B-Cell Lymphomas: Results of a Phase 1b-II Study of the LYSA Group

**DOI:** 10.3390/cancers14071761

**Published:** 2022-03-30

**Authors:** Christophe Bonnet, Jehan Dupuis, Hervé Tilly, Thierry Lamy, Christophe Fruchart, Steven le Gouill, Catherine Thieblemont, Franck Morschhauser, Olivier Casasnovas, Krimo Bouabdallah, Hervé Ghesquieres, Eric Van Den Neste, Marc André, Guillaume Cartron, Gilles Salles

**Affiliations:** 1Clinical Hematology Unit, Centre Hospitalier Universitaire de Liège, Liège Université, B35, Campus Universitaire du Sart-Tilman, 4000 Liège, Belgium; 2Lymphoid Malignancies Unit, Hôpital Henri Mondor, 51 Avenue du Maréchal de Lattre de Tassigny, CEDEX, 94000 Créteil, France; jehan.dupuis@aphp.fr; 3Hematology Department Henri Becquerel, Rue d’Amiens, CEDEX, 76000 Rouen, France; herve.tilly@chb.unicancer.fr; 4Centre Hospitalier Universitaire de Rennes, Service d’Hématologie Clinique, 35033 Rennes, France; thierry.lamy-de-la-chapelle@univ-rennes1.fr; 5Institut National de la Recherche Médicale, Unité Mixte de Recherche U1236, Université de Rennes 1, Etablissement Français du Sang Bretagne, 35000 Rennes, France; 6Hematology Unit, Centre Hospitalier de Dunkerque, 130, Avenue Louis Herbeaux, CS 76 367, CEDEX 1, 59385 Dunkerque, France; christophe.fruchart@ch-dunkerque.fr; 7Institut Curie, 26 Rue d’Ulm, 75005 Paris, France; steven.legouill@curie.fr; 8Hemato-Oncology Department, Hôpital Saint Louis, 1 Av. Claude Vellefaux, 75010 Paris, France; catherine.thieblemont@aphp.fr; 9Hematology Unit, Centre Hospitalier de Lille, 2, Avenue Oscar Lambret, 59037 Lille, France; franck.morschhauser@chru-lille.fr; 10Department of Hematology, Centre Hospitalier Universitaire de Dijon Bourgogne and INSERM 1231, 14 Rue Paul Gaffarel, 21000 Dijon, France; olivier.casasnovas@chu-dijon.fr; 11Hematology Department, Centre Hospitalier de Bordeaux, 12, Rue Dubernat, 33000 Bordeaux, France; krimo.bouabdallah@chu-bordeaux.fr; 12Hematology Unit, Hopital Lyon Sud, Pierre-Benite, 165 Chemin du Grand Revoyet, 69310 Pierre-Bénite, France; herve.ghesquieres@chu-lyon.fr; 13Hematology Unit, Cliniques Universitaires Saint-Luc, 10 Avenue Hippocrate, 1200 Bruxelles, Belgium; eric.vandenneste@saintluc.uclouvain.be; 14Department of Hematology, Centre Hospitalier Universaitaire, Université Catholique de Louvain Namur, Université Catholique de Louvain, 1 Rue Dr Gaston Therasse, 5530 Yvoir, Belgium; marc.andre@chuuclnamur.uclouvain.be; 15Department of Hematology, Centre Hospitalier Universtaire de Montpellier, UMR-CNRS 5535, 34090 Montpellier, France; g-cartron@chu-montpellier.fr; 16Department of Medicine, Memorial Sloan Kettering Cancer Center, Weill Cornell Medical College, 530 East 74th Street, New York, NY 10065, USA; sallesg@mskcc.org

**Keywords:** relapsed/refractory non-Hodgkin B-cell lymphoma, ibrutinib, R-DHAP, R-DHAOx, safety

## Abstract

**Simple Summary:**

Patients with relapsing/refractory B-cell lymphoma who respond to a platinum salt-based salvage regimen can be cured after therapy intensification followed by autologous stem cell transplantation. The Bruton tyrosine kinase inhibitor ibrutinib, given alone or in association with other molecules, has proven effective in numerous B-cell lymphomas. The aim of the current study was to evaluate the safety of the combination of ibrutinib, rituximab, dexamethasone, and cytarabine with either cisplatin (R-DHAP) or oxaliplatin (R-DHAOx), with ibrutinib given based on a prespecified dose-escalation design. The concomitant combination of ibrutinib and R-DHAP resulted in limiting hematological, infectious, and renal toxicities. As a result, the maximum ibrutinib dose could not be administered. On the other hand, when the ibrutinib administration schedule was combined with R-DHAOx, ibrutinib dosing could be increased up to the maximum prespecified dose but with relevant toxicities. Despite a strong rationale for combining ibrutinib with either R-DHAP or R-DHAP/Ox, this approach was limited by significant toxicities.

**Abstract:**

In the post-rituximab era, patients with relapsed/refractory non-Hodgkin B-cell lymphoma (R/R B-NHL) responding to a platinum salt-based salvage regimen can potentially be cured after intensification followed by autologous stem cell transplantation, with the quality of the response to salvage predicting survival. The Bruton tyrosine kinase inhibitor ibrutinib, given as monotherapy or combined with other molecules, has proven effective in numerous B-cell lymphomas. To evaluate the safety of the combination of ibrutinib, rituximab, dexamethasone, and cytarabine with either cisplatin (R-DHAP) or oxaliplatin (R-DHAOx), we conducted a multicenter Phase 1b-II study in transplant-eligible R/R B-NHL patients, with ibrutinib given using a 3-by-3 dose-escalation design. The combination of R-DHAP and ibrutinib (given from Day 1 to Day 21 of each cycle) was associated with dose-limiting hematological, infectious, and renal toxicities, while we were unable to reach a dose to recommend for Phase II. R-DHAOx could only be combined with a daily dosage of 280 mg ibrutinib when administered continuously. R-DHAP combined with intermittent ibrutinib administration (from Day 5 to Day 18) was found to be highly toxic. On the other hand, when this administration schedule was combined with R-DHAOx, ibrutinib dosing could be increased up to 560 mg but with relevant toxicities. Despite a strong rationale for combining ibrutinib and R-DHAP/R-DHAOx, as both target lymphoma B-cells by different mechanisms, this approach was limited by significant toxicities.

## 1. Introduction

Diffuse large B-cell lymphoma (DLBCL) is the most common form of non-Hodgkin lymphoma, with a 50–60% cure rate following anthracycline-based immunochemotherapy [1]. For patients who relapse or are refractory, autologous stem cell transplantation (ASCT) with curative intent has remained the standard of care for patients eligible for this procedure. Nevertheless, only half of the eligible patients are able to proceed to transplant due to failure of salvage, while among responding patients, those unable to reach a complete response after salvage have a significant risk of relapse after ASCT [2,3]. Patients with second-line treatment or relapse after ASCT now have access to CAR-T cell therapy [4,5]. Besides CAR-T, bispecific T-cell engager, by increasing lymphocyte cytotoxic activity, offers new intriguing opportunities [6,7]. A salvage regimen prior to ASCT should enable tumor reduction with acceptable safety before proceeding with one of those procedures. Therefore, new strategies to improve the efficiency of salvage regimens are warranted.

Typical salvage regimens preceding ASCT comprise platinum-based immunochemotherapies, such as rituximab, dexamethasone, cytarabine, and either cisplatin (R-DHAP) or oxaliplatin (R-DHAOx); a combination of ifosfamide, etoposide, and carboplatin (R-ICE); and combinations of rituximab, gemcitabine, dexamethasone, and cisplatin (R-GDP) [7,8]. New strategies to improve the proportion of ASCT-eligible patients are urgently needed [8,9,10]. In the lymphoma field, more than 20 new agents are currently undergoing clinical evaluation, given either alone or in combination, with the aim to improve salvage regimen response rates [4].

Among them, ibrutinib is the first-in-class irreversible Bruton tyrosine kinase (BTK) inhibitor that induces apoptosis in lymphoma cells. Ibrutinib has shown an overall 37% response rate in DLBCL of the activated B-cell subtype, and the agent is active in other B-cell malignancies but with low complete response (CR) rates when given as a single agent. Its metabolism in the liver is decreased in the event of its concomitant use with CYP3A4 inhibitors, but it is not affected by renal insufficiency [11]. Because of its safety profile, some combinations with immunochemotherapy were rendered feasible, including R-CHOP and R-ICE regimens, with efficacy data available in young patients [12,13]. Data in ovarian cancer suggest synergic effects of cisplatin and ibrutinib [14]. In CLL, Waldenström’s macroglobulinemia, and MCL, the combination of ibrutinib, rituximab, and bendamustin was associated with improved response and survival rates [15,16,17]. Overall, these results constitute a sufficiently strong rationale to envision that the quality of treatment responses before transplantation in relapsed/refractory non-Hodgkin B-cell lymphoma (R/R B-NHL) patients can be enhanced by conducting a Phase 1b-II study to further evaluate ibrutinib given in combination with either R-DHAP or R-DHAOx.

## 2. Materials and Methods

### 2.1. Study Design and Patients

We performed an open-label Phase 1b, non-randomized, dose-escalation study of ibrutinib in combination with platinum-based regimens (Phase 1b, 50 patients), which was followed by a single expansion cohort across 15 centers of the Lymphoma Study Association (LYSA) in France and Belgium. In this parallel study, two regimens were assessed, R-DHAP and R-DHAOx, as Groups A and B, respectively.

Inclusion criteria were patients with confirmed B-cell lymphoma (DLBCL and other large B-cell, follicular, mantle cell, nodal marginal zone, and transformed indolent lymphomas) with relapsed or refractory disease after up to two therapy lines, aged between 18 and 70 years, and fit for ASCT. Other eligibility criteria included an Eastern Cooperative Oncology Group (ECOG) performance status of 2 or less and at least one measurable lymphoma lesion based on the 2007 revised response criteria for malignant lymphoma [18].

All biopsies were centrally reviewed to confirm diagnosis. DLBCL cases were classified using the Hans algorithm as germinal center (GC) or non-germinal center (non-GC) immunophenotypes.

Exclusion criteria included previous treatment with any BTK inhibitor; progression or refractoriness after treatment with any phosphoinositide 3-kinase (PI3K) inhibitor; history of stroke or intracranial hemorrhage within the 6 months before inclusion or major surgery within 3 weeks before enrollment; known bleeding diathesis, platelet function disorders and need for therapeutic anticoagulation (i.e., aspirin, low-molecular-weight heparin, or any other anticoagulants); treatment with strong CYP3A/4 inhibitors; central nervous system involvement; history of human immunodeficiency virus or active hepatitis B or C virus infection; left ventricular ejection fraction < 45% or any relevant cardiovascular disease; altered liver or renal functions; preexisting Grade ≥2 neuropathy.

Patients were scheduled to receive three cycles of rituximab and high-dose cytarabine with either cisplatin (R-DHAP) [19] or oxaliplatin (R-DHAOx) [20], which were designated as Groups A and B, respectively. Allocation of each patient to either Group A (cisplatin) or Group B (oxaliplatin) was left to the investigator’s discretion. Cycles were to be administered every 21 days, with continuous concomitant daily oral ibrutinib administration starting with Dose Level 1 at 420 mg daily. A dose-escalation committee (DEC), consisting of the coordinator, co-coordinator, principal investigators, and sponsor’s medical monitors, reviewed Cycle 1 data to determine dose-limiting toxicities (DLTs), on the basis of which the dose was escalated to Dose Level 2 (560 mg daily) or de-escalated to Dose Level 1 (280 mg daily).

Due to the combination’s excessive toxicities (see below), the protocol was amended to perform a second parallel dose-escalation cohort, named “Day 5–18 dose-escalation cohort”, consisting of an intermittent daily administration of ibrutinib from Day 5 until Day 18 of each cycle. Subsequently, owing to the observed renal toxicity of the R-DHAP and intermittent ibrutinib combination, the protocol was amended to close the R-DHAP cohort and enroll patients only into the R-DHAOx and “D5–D18” ibrutinib expansion cohort (25 patients). Finally, the emergence of other safety signals resulted in trial discontinuation on 23 March 2018, after enrollment of 75 patients (see below).

### 2.2. Procedures

For each dose-escalation cohort, the standard 3 + 3 scheme was applied to define the maximum tolerated dose (MTD) of ibrutinib in combination with immunochemotherapy. Ibrutinib (Pharmacyclics LLC, an Abbvie Company, South San Francisco, CA, USA) was taken either continuously (daily) from Day 1 to Day 21 (first dose escalation, Groups A and B) or intermittently from Day 5 to Day 18 (“D5–D18” escalation cohorts, Groups A’ and B’ and expansion Group B’), with 21-day cycles of R-DHAP/R-DHAOx (Groups A and B, respectively) (intravenous rituximab 375 mg/m² on Day 1, dexamethasone 40 mg total dose on Days 1 to 4, cytarabine 2 g/m² given every 12 h on Day 2, and cisplatin 100 mg/m² or oxaliplatin 130 mg/m² on Day 1). Dose escalation started with ibrutinib 420 mg/day (Dose Level 1) with the possibility to escalate to 560 mg or de-escalate to 280 mg/day. Dose re-escalation of ibrutinib was not allowed. DLTs were recorded during the first cycle of immunochemotherapy.

All patients received mandatory primary infectious prophylaxis by granulocyte grow factor injections (filgrastim or pegfilgrastim) and were required to receive prophylactic antibiotics (sulfamethoxazole/trimethoprim) and antiviral therapy (valaciclovir) according to local practice at each center.

Patients were assessed clinically on Day 1 of each cycle. Adverse events (AEs) were recorded from screening up to 30 days after the last ibrutinib intake and graded using the Common Terminology Criteria for Adverse Events (CTCAE) Version 4.0. Serious adverse events (SAEs) were defined as those that resulted in death, admission to hospital, persistent or significant disability, or life-threatening or medically significant events. Adverse events of special interest (AESIs) comprised any intracranial or other Grade ≥ 3 hemorrhage. Dose adjustments for ibrutinib in the event of AESI were provided. In case of hematological Grade ≥ 3 toxicity that was potentially attributable to ibrutinib, the drug could be interrupted for up to 2 weeks. In this case, the drug could be restarted at a lower dose. Dose reductions of 25% were recommended in the case of Grade >3 hematological toxicities lasting more than 7 days (any cytotoxic drug), Grade ≥ 2 neuropathy (cisplatin and oxaliplatin), Grade ≥ 2 nephropathy (cisplatin), or Grade ≥ 3 hematological toxicity lasting more than 7 days (cytarabine). DLTs were defined as the occurrence of any Grade ≥3 non-hematological toxicity (except for alopecia, Grade 3 diarrhea, and vomiting or asthenia over less than 7 days) and Grade 4 hematological toxicities (except for lymphopenia) over more than 7 days.

The study (NCT02055924) was sponsored by the Lymphoma Academic Research Organization (LYSARC) and conducted according to the Declaration of Helsinki and the International Conference on Harmonization Guidelines for Good Clinical Practice. The protocol was approved by the Comité de Protection des Personnes (Ethics Committee) Sud-Est II, Lyon, in France and the Comité d’Ethique in Belgium. All patients provided written informed consent.

### 2.3. Endpoints

The data of patients treated with either R-DHAP or R-DHAOx were analyzed separately. The primary endpoint of Phase 1b was the determination of the recommended Phase II dose (RP2D) of ibrutinib in combination with R-DHAP/R-DHAOx (DLT Set).

Secondary endpoints comprised the recording of safety data (all AEs occurring in patients who received at least one ibrutinib dose) and efficacy signals (CR or partial response (PR)) at the end of the three combination cycles, the duration of response, and progression-free (PFS) and overall survival (OS). Response was evaluated within 30 days after intake of the last ibrutinib dose, before the high-dose therapy and ASCT procedure or the start of any alternate anti-lymphoma therapy, using 2007 Cheson criteria [18]. During follow-up after ASCT completion, patients were clinically assessed every 3 months, with chest, abdomen, and pelvic CTs performed every 6 months. Patients experiencing lymphoma progression or developing unacceptable toxicity were withdrawn from the study.

Exploratory endpoints comprised the feasibility of stem cell collection and, for DLBCL patients, correlation between cell of origin (assessed by IHC) and response.

### 2.4. Statistical Analyses

For the escalation cohort, the total number of patients enrolled was defined using a 3 + 3 design. For the expansion cohort, all analyses were exploratory in nature, without any specific hypothesis testing or formal power calculations performed.

Safety analyses were conducted on the safety set and summarized descriptively. The main criterion analysis was conducted on the DLT set. The safety set consisted of all enrolled patients who took at least one ibrutinib dose. The DLT set included all patients enrolled in the escalation cohort who received full immunochemotherapy doses and ≥80% of the total ibrutinib dose during Cycle 1, unless the missed doses were due to study drug-related AEs. The efficacy set included all patients who received at least 75% of the total ibrutinib dose during Cycle 1. Overall, 95% confidence limits of response rates were calculated according to the exact Pearson–Clopper method. Survival functions were estimated using the Kaplan–Meier method with appropriate 95% CIs.

## 3. Results

Between 26 May 2014 and 23 March 2018, a total of 75 patients were enrolled in the study. The first 25 were allocated to receive concomitant administration of ibrutinib (13 with R-DHAP: Group A; 12 with R-DHAOx: Group B) in the first dose-escalation cohorts, and 50 received intermittent ibrutinib administration, including 25 in intermittent dose-escalation cohorts (13 with R-DHAP: Group A’; 12 with R-DHAOx: Group B’) and 25 in the subsequent intermittent expansion cohort, all in combination with R-DHAOx (Figure 1).

Two-thirds of enrolled patients were male, 47% over 60 years of age, and the majority displayed advanced-stage disease (Table 1). Eight (11%) patients exhibited an ECOG performance status of 2. The most common lymphoma entities were DLBCLs (N = 30, 41%) and follicular lymphomas (FLs) (N = 19, 26%). The median number of previous treatment lines was one, with only eight patients having received two or more previous therapy lines. A greater proportion presented with disease responsive to previous therapy (Group A: 77%; Group B: 83%; Group B’: 64%), except for Group A’, in which 50% had refractory disease, defined as progressive disease during either first-line treatment or within 6 months of completing first-line treatment.

### 3.1. Safety and Tolerability

Overall, 25 patients were treated with R-DHAP, and 49 were treated with the R-DHAOx and ibrutinib combination.

#### 3.1.1. Ibrutinib in Combination with R-DHAP

The 25 patients who received R-DHAP in combination with ibrutinib were evaluated for safety (safety set; one patient was excluded due to renal insufficiency during Cycle 1 chemotherapy) (Figure 1). Among them, 13 received concomitant (Group A) ibrutinib administration, and 12 received intermittent (“D5–D18”, Group A’) ibrutinib administration. All patients developed at least one or more Grade 3 AEs (Table 2). The most common AEs included cytopenias (Groups A and A’: 100%), renal failures (Group A: 54%; Group A’: 67%), infections (Group A: 31%; Group A’: 25%, without any case of atypical pneumonia), and metabolic, gastro-intestinal, and cardiac toxicities. In total, 20 patients developed at least one SAE, the most common being renal failure (*n* = 12 in 12 patients, 48%) and infections (*n* = 6 in 5 patients, 20%). One Group A patient developed a Grade 4 cutaneous eruption. Treatment interruption occurred in seven patients (including the death of one Group A patient, which occurred consequently to multiple organ failure due to sepsis developed after prolonged neutropenia). Out of 25 patients included in the DLT set, 8 (35%) (5 in Group A and 3 in Group A’) experienced 13 DLTs (Table 3). In Group A, three DLTs occurred among the six patients treated with ibrutinib 420 mg (Dose Level 1), and four DLTs occurred (in two patients) among the seven subsequent patients treated with ibrutinib 280 mg (Level 1). In Group A’ (intermittent D5–D18 dosing), no DLTs occurred in four patients receiving ibrutinib at Dose Level 1 (420 mg), whereas three out of the six patients treated at Dose Level 2 (560 mg) developed six DLTs. Given the pattern of toxicities still occurring at Dose Level 1 (420 mg), the decision was made not to pursue the investigation of this combination using DHAP.

#### 3.1.2. Ibrutinib in Combination with R-DHAOx

Overall, 48 out of the 49 patients enrolled in Groups B and B’ (R-DHAOx and concomitant ibrutinib or “D5–D18” intermittent ibrutinib, respectively) were included in the safety analysis (Group B: 12 patients; Group B’: 36 patients; 1 patient was withdrawn from the expansion cohort because of symptomatic coronary syndrome before ibrutinib administration) (Figure 1). In total, 45 patients developed at least one AE (Table 2), with cytopenia (Group B: 100%; Group B’: 92%), infections (Group B: 33%; Group B’: 22%), and neurological (Group B: 25%; Group B’: 22%), metabolic (Group B: 17%; Group B’: 17%), hepato-biliary (Group B: 17%; Group B’: 14%), gastro-intestinal (Group B’: 17%), and cardiac toxicities (Group B: 11%) being the most common ones. SAEs were recorded in 23 patients, including infections (*n* = 11), cutaneous eruptions (*n* = 2, both after continuous administration of ibrutinib), atrial fibrillation (*n* = 2), hemorrhagic events (*n* = 2), and veno-occlusive disease (VOD, *n* = 3; occurring during ASCT). Two patients prematurely interrupted the study because of AEs (allergy to oxaliplatin and Grade 3 epigastric pain).

Six DLTs (one Grade 4 thrombocytopenia, one Grade 3 febrile neutropenia, two Grade 3 cutaneous eruptions, one Grade 3 prostatic infection, and one Grade 3 epigastric pain; five with concomitant administration and one with “D5–D18” ibrutinib administration) were observed in five patients (Table 3). Five DLTs occurred in three out of the six patients treated with continuous concomitant ibrutinib at Dose Level 1 (420 mg, Group B). One of the six subsequent patients treated at Dose Level 1 (280 mg) developed Grade 3 epigastric pain. Though the 280 mg daily dose of continuous ibrutinib was eventually deemed safe, the DEC decided not to pursue an expansion cohort, as this dose level might provide only suboptimal BTK inhibition in lymph nodes and across a range of individual body weights [4].

Among the six patients receiving 420 mg “D5–D18” ibrutinib treatment (Dose Level 1, Group B’), one developed Grade 3 hepatitis and then liver veno-occlusive disease after ASCT. None of the six subsequent patients treated at Dose Level 2 (560 mg) developed any DLTs. For this group of patients receiving “D5–D18” ibrutinib administration, a recommended Phase II dose of 560 mg ibrutinib, given from Day 5 to 18, was selected for the expansion cohort.

### 3.2. Patient Outcomes

Multiple safety issues and early withdrawal observed in patients receiving ibrutinib in combination with R-DHAP (Groups A and A’) or continuously with R-DHAOx (Group B) precluded the interpretation of the efficacy of these combination regimens. Hence, outcome endpoints were not further evaluated for these cohorts. In our analysis, we thus focused on all patients in Group B’ receiving “D5–D18” ibrutinib with R-DHAOx (35 patients, efficacy set; see study flowchart, Figure 1), 33 (94%) of them having received the complete treatment in Cycle 1 and 30 receiving it in all planned cycles. At the end of treatment, overall response (OR) was observed in 26 patients (74.3%, 95%CI: 56.7–87.5%), including 17 (48.6%, 95%CI: 31.4–66.0%) achieving a CR across all histology types. For DLBCL patients that were evaluable for COO status based on the Hans algorithm (*n* = 16), OR and CR rates were respectively 90% (95%CI: 55.5–99.7%) and 60.0% (95%CI: 26.2–87.8%) for non-GC profile patients (N = 10) and 50.0% (95%CI: 11.8–88.2%) (both for OR and CR) for GC profile patients (N = 6).

The median PFS was 12.2 months (95%CI: 4.5–not reached) for all 35 Group B’ patients and 16.6 months (95%CI: 2.6–not reached) for those with DLBCL (Figure 2). At a median follow-up of 14.1 months (95%CI: 12.0–23.2), 10 (28%) patients died, all after completing ibrutinib–R-DHAOx combination therapy, with 7 deaths related to lymphoma, 2 during the ASCT procedure (1 with veno-occlusive disease and 1 with organ toxicity during the conditioning regimen), and 1 patient upon CLL progression (unrecognized at the time of study inclusion). The 12-month OS was 74.3% for the 35 Group B’ patients.

Out of the 35 mobilized patients, 4 patients did not reach sufficient progenitor cell collection yield to proceed to ASCT. A median of 5.29 × 10^6^ (0.90; 35.72) CD34+ cells were obtained in the remaining 31 patients after one (24 patients), two (5 patients), or three (2 patients) apheresis procedures.

## 4. Discussion

The current results indicate that the administration of ibrutinib with the continuous daily dosing of either R-DHAP or R-DHAOx results in significant toxicities that preclude their further administration as salvage regimens before transplant.

Salvage chemotherapy followed by ASCT has been the standard of care for patients with refractory/relapsing DLBCL after first-line therapy, although CAR-T therapy likely provides additional benefits in certain categories of patients [21,22]. When used before intensification, these regimens should be efficient in achieving optimal responses, but they must also be safe so as to maintain patients in a sufficiently good general condition, thereby enabling them to proceed to ASCT. Randomized studies have compared R-DHAP versus either R-ICE or R-GDP, with comparable efficacy results observed [8,9]. In the Coral study, hematological toxicities were more common with the R-DHAP regimen, resulting in increased platelet transfusion requirements, as compared to R-ICE (57% of patients receiving platelet transfusions during Cycle 1 in the R-DHAP arm versus 35% in the R-ICE arm). Notably, our group previously reported an inferior outcome for patients with non-GC profile DLBCL compared to GCB at the time of salvage, along with the superiority of R-DHAP versus R-ICE for patients with a GC profile [23].

Due to its more favorable safety profile with respect to renal function, cisplatin is frequently replaced by oxaliplatin, although no randomized study has demonstrated equivalent efficacy so far [8,20,24,25].

At the time of study design, the preliminary results of the early-phase study combining ibrutinib (420 mg daily) and R-CHOP were encouraging, particularly for patients with non-GC DLBCL [12]. However, the PHOENIX randomized study comparing R-CHOP with or without ibrutinib for untreated non-germinal center DLBCL patients did not demonstrate an OS benefit with this combination, except for in patients under 60 years of age, suggesting excessive toxicity in older patients, limiting the potential benefit of this combination. It must be noted that recent data indicate that some molecularly defined subgroups of DLBCL patients could benefit from this combination [26,27].

The combination of continuous ibrutinib administration and R-DHAP was associated with significant hematological toxicities, prohibiting ibrutinib escalation to Dose Level 2 (560 mg). Despite the dose reduction to 280 mg, hematological toxicity remained limiting, and no recommended Phase II dose could be identified for patients receiving R-DHAP.

The combination of continuous 420 mg ibrutinib and R-DHAOx similarly caused relevant hematological toxicities necessitating the de-escalation of ibrutinib to Dose Level 1, potentially preventing optimal exposure of lymphoma cells to this agent. While daily doses of ibrutinib above 2.5 mg/kg have shown over 95% of receptor occupancy in peripheral blood [4], data on BTK inhibitors suggest that BTK occupancy in lymph nodes requires optimal exposure to these agents [28].

Suspecting that pharmacokinetic modifications due to a pharmacological interaction between ibrutinib and chemotherapeutic agents explained the high rate of cutaneous eruptions and hematological toxicities, the study was amended with intermittent (D5–D18 administration in each cycle) rather than continuous administration of ibrutinib. However, even with this intermittent schedule, patients treated with R-DHAP developed prohibitive renal toxicities (eight SAEs in eight patients), leading to the discontinuation of the development of this combination. Administration of R-DHAOx and intermittent ibrutinib (Group B’) appeared to be safer, and it was possible to reach a recommended Phase II dosage of 560 mg. However, in addition to the high rates of hematological toxicities (Grades 3 and 4 in 96% of patients) that we encountered, during the expansion cohort, limiting infectious complications occurred, as well (10 SAEs in 6 patients). In addition, three episodes of liver veno-occlusive disease in three patients were observed during the following SCT. This thrombotic microangiopathy has previously been described following oxaliplatin administration without ibrutinib and eventually during ASCT, with a mean incidence estimated at around 14% [29,30].

Regarding AESIs, seven (10%) patients developed atrial fibrillation, all with a favorable outcome, which is the expected incidence after ibrutinib exposure. Despite severe hematological toxicities (Grade > 2 thrombocytopenia recorded in 75 up to 100% of patients, depending upon the immunochemotherapy associated with ibrutinib), only three hemorrhagic complications were recorded on account of efficient platelet transfusion support.

Considering the efficacy in the 16 DLBCL patients treated with 560 mg of intermittent ibrutinib and R-DHAOx, the 90% ORR and 60% CRR of the 10 patients with a non-GC profile appear quite encouraging.

In the randomized PHOENIX Phase III study, prolonged neutropenia with more infectious complications in the patient group receiving ibrutinib was observed, as well, in line with our findings [27]. In a Phase II randomized study, Canadian investigators assessed the combination of ibrutinib with rituximab/gemcitabine/dexamethasone/cisplatin (GDP) [31]. In that study, 14 DLBCL patients receiving continuous ibrutinib administration at a daily dosage of 560 mg displayed a lower ORR than those treated without ibrutinib, while 5 severe infectious events occurred (with one Grade 5 sepsis and one Grade 5 pneumonia). The combination of R-ICE with continuous ibrutinib given at the daily dosage of 840 mg resulted in encouraging anti-tumor effects in patients with non-GC DLBCL. Although a different platin salt (carboplatin) is part of this regimen, infectious toxicity remained significant in that study, with 4 episodes of febrile neutropenia in 4 of the 21 patients and 4 episodes of Grade 3 infections in 4 others [13].

## 5. Conclusions

In conclusion, the combination of ibrutinib and platinum-based immunochemotherapy in our trial was associated with prohibitive hematological and infectious complications, thereby significantly limiting ibrutinib exposure and potentially blunting any potential benefit of this combination. CAR-T cell therapies, as well as bispecific antibodies in patients with relapsed or refractory B-cell lymphoma, also appear promising (although the latter had a short follow-up) [32,33]. How targeted therapies can be safely combined with platinum salt-containing salvage regimens in order to improve their efficacy is thus still an open question, which must be further investigated.

## Figures and Tables

**Figure 1 cancers-14-01761-f001:**
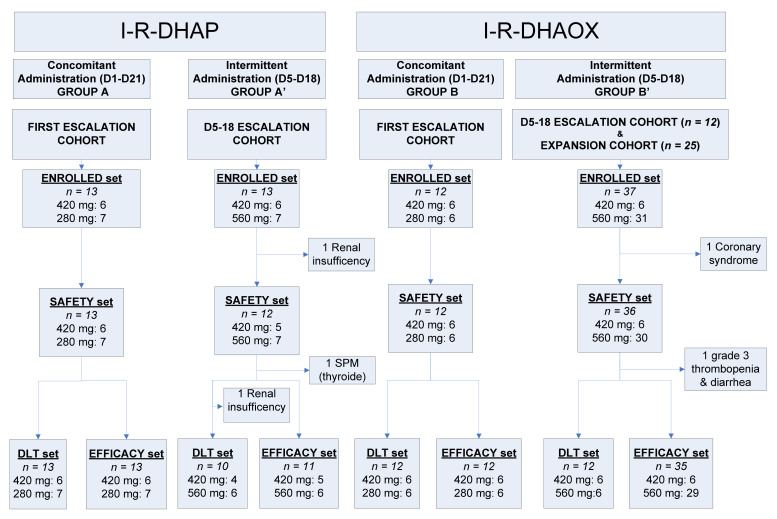
Study flowchart. D: day; DLT: dose-limiting toxicity; I–R–DHAOx: ibrutinib, rituximab, dexamethasone, cytarabine, and oxaliplatin; I–R–DHAP: ibrutinib, rituximab, dexamethasone, cytarabine, and cisplatin.

**Figure 2 cancers-14-01761-f002:**
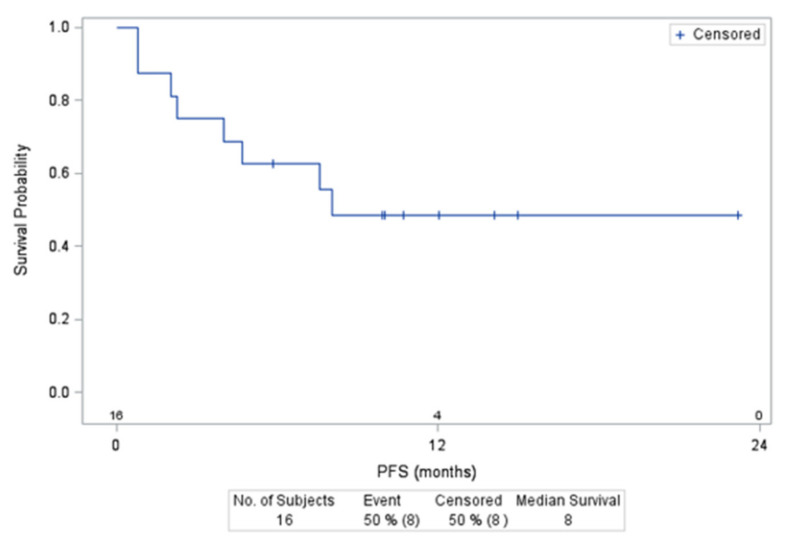
Progression Free Survival (PFS) of DLBCLpatients included in the group B’ treated with 560 mg of ibrutinib.

**Table 1 cancers-14-01761-t001:** Patient characteristics (enrolled and safety sets).

Type of Immunochemotherapy	R-DHAP		R-DHAOx	
Administration of Ibrutinib	Concomitant	Intermittent	Concomitant	Intermittent
	D1–21	D5–18	D1–21	D5–18
Group	A	A’	B	B’
Patients enrolled	13	13	12	37
Patients evaluable for DLT	13	10	12	12
Patients evaluable for safety	13	12 ^£^	12	36 ^££^
Median age *	63 (40–68)	60 (49–69)	58 (42–66)	62 (25–70)
≤60 years	5 (38%)	9 (75%)	8 (67%)	16 (44%)
>60 years	8 (62%)	3 (25%)	4 (33%)	20 (56%)
Sex				
Male	9 (69%)	8 (67%)	8 (67%)	23 (64%)
Female	4 (31%)	4 (33%)	4 (33%)	13 (36%)
Stage at inclusion				
I	2 (15%)	4 (33%)	3 (25%)	3 (8%)
II	3 (23%)	1 (8%)	0	3 (8%)
III	2 (15%)	3 (25%)	2 (17%)	6 (17%)
IV	6 (45%)	4 (33%)	7 (58%)	24 (67%)
ECOG at inclusion				
0	8 (63%)	9 (75%)	8 (67%)	25 (69%)
1	3 (23%)	3 (25%)	3 (25%)	9 (25%)
2	2 (15%)	0	1 (8%)	2 (6%)
LDH > upper normal limit				
No	6 (69%)	7 (64%)	3 (75%)	18 (50%)
Yes	7 (31%)	4 (36%)	9 (25%)	18 (50%)
Bone marrow				
Performed	11 (85%)	9 (75%)	11 (92%)	34 (94%)
Involved	2 (18%)	0	2 (18%)	6 (79%)
Non-involved	9 (82%)	9 (100%)	8 (73%)	28 (82%)
Unspecified	0	0	1 (9%)	0
Histology at inclusion				
DLBCL	4 (31%)	4 (33%)	7 (58%)	15 (46%)
Other large B-cell lymphoma ^†^	0	2 (16%)	1 (8%)	2 (6%)
Follicular lymphoma Grade 1, 2, or 3a	2 (15%)	4 (33%)	2 (16%)	11 (33%)
Nodal marginal zone lymphoma	1 (8%)	0	1 (8%)	1 (3%)
Mantle cell lymphoma	2 (15%)	0	1 (8%)	0
Transformed indolent lymphoma ^‡^	4 (32%)	1 (8%)	0	3 (9%)
Missing	0	2 (17%)	0	4 (13%)
Disease status				
Refractory	3 (23%)	6 (50%)	3 (25%)	13 (36%)
Relapsed/progressive	10 (77%)	6 (50%)	9 (75%)	23 (64%)
Number of previous treatment lines				
1	12 (92%)	12 (100%)	10 (83%)	31 (86%)
2	1 (8%)	0	2 (17%)	5 (14%)
Previous treatments				
Rituximab	13 (100%)	11 (92%)	10 (83%)	36 (100%)
Anthracycline-based chemotherapy	13 (100%)	11 (92%)	12 (100%)	35 (97%)
Radiotherapy	1 (8%)	0	3 (25%)	1 (3%)
Autologous stem cell transplantation	2 (15%)	0	0	1 (3%)
Allogenic stem cell transplantation	0	0	1 (8%)	0

D: day; DLBCL: diffuse large B-cell lymphoma; DLT: dose-limiting toxicity; ECOG: Eastern Cooperative Oncology Group; LDH: lactate dehydrogenase; R-DHAOx: rituximab, dexamethasone, cytarabine, and oxaliplatin; R-DHAP: rituximab, dexamethasone, cytarabine, and cisplatin. £: One patient did not receive any ibrutinib due to renal failure on Day 3; ££: One patient did not receive any ibrutinib due to acute coronary syndrome on Day 3; †: Mediastinal large B-cell lymphoma, diffuse large B-cell lymphoma plasmablastic, or diffuse large B-cell lymphoma unclassifiable; ‡: Association of diffuse large B-cell lymphoma and follicular lymphoma, nodal marginal zone lymphoma, chronic lymphocytic leukemia, or small B-cell lymphoma. All data are N (%) except *: N (min–max).

**Table 2 cancers-14-01761-t002:** Ibrutinib combined with R-DHAP/R-DHAOx—toxicities.

Chemotherapy	R-DHAP				R-DHAOx			
Enrolled set	26				49			
Safety set	25				48			
**Ibrutinib administration**	**Concomitant**		**Intermittent**		**Concomitant**		**Intermittent**	
Group	A		A’		B		B’	
Enrolled set	13		13		12		37	
Safety set	13		12		12		36	
**Grade >2 AEs in >10% of patients and AESIs**	**Patients (%)**	**AE**	**Patients (%)**	**AE**	**Patients (%)**	**AE**	**Patients (%)**	**AE**
	**13 (100%)**	**127**	**12 (100%)**	**98**	**12 (100%)**	**81**	**33 (92%)**	**171**
Blood and lymphatic disorders	13 (100%)	87	12 (100%)	56	12 (100)%	59	30 (83%)	111
Thrombocytopenia	13 (100%)	31	9 (75%)	18	12 (100%)	26	30 (83%)	65
Neutropenia	11 (85%)	18	6 (50%)	9	5 (42%)	8	12 (33%)	17
Febrile neutropenia	6 (47%)	8	3 (25%)	3	3 (25%)	3	3 (8%)	4
Anemia	8 (62%)	11	3 (25%)	9	2 (17%)	4	8 (22%)	9
Infections	4 (31%)	4	3 (25%)	5	4 (33%)	4	8 (22%)	12
Bacterial sepsis	2 (15%)	2						
Bronchitis			2 (17%)	3				
Hemorrhagic manifestations	2 (15%)	2					1 (3%)	1
General disorders and administration site conditions	2 (15%)	3	2 (17%)	2			3 (8%)	3
Cardiac disorders	2 (15%)	4	3 (25%)	3			4 (11%)	5
Atrial fibrillation	2 (15%)	2	2 (17%)	2			3 (8%)	3
Vascular disorders	1 (8%)	1	1 (8%)	1				
Metabolism and nutrition disorders	6 (46%)	9	1 (8%)	1	2 (17%)	2	6 (17%)	11
Hypocalcemia	3 (23%)	3						
Hypokalemia	2 (15%)	2			1 (8%)	1	2 (6%)	3
Renal and urinary disorders	7 (54%)	7	8 (67%)	8				
Renal failure	7 (54%)	7	8 (67%)	8				
Gastrointestinal disorders	4 (31%)	5	2 (17%)	3			6 (17%)	7
Diarrhea							4 (11%)	4
Hepatobiliary disorders					2 (17%)	2	5 (14%)	7
Congenital, familial, and genetic disorders			2 (17%)	2				
Aplasia			2 (17%)	2				
Nervous system disorders					3 (25%)	3	8 (22%)	10
Peripheral neuropathy					1 (8%)	1	7 (19%)	7
Skin and subcutaneous disorders	2 (15%)	2			2 (17%)	2		
Second primary neoplasias			2 (17%)	2				
**Patients with at least one SAE**	**Patients (%)**	**SAE**	**Patients (%)**	**SAE**	**Patients (%)**	**SAE**	**Patients (%)**	**SAE**
	**10 (77)**	**27**	**10 (77%)**	**34**	**6 (50%)**	**17**	**17 (47%)**	**30**
Infection	3 (23%)	4	2 (17%)	2	5 (42%)	6	6 (27%)	10
Renal failure	4 (31%)	4	8 (67%)	8				
Atrial fibrillation	1 (8%)	1	1 (8%)	1			2 (6%)	2
Cutaneous eruption	1 (8%)	1			2 (17%)	2		
Hemorrhagic complication	2 (17%)	2					2 (6%)	2
Veno-occlusive disease							3 (8%)	3
**DLT evaluable patients (*n*)**	**13**		**10**		**12**		**12**	
Patients with DLT	5		3		4		1	
Number of DLTs	7		6		6		1	
**Patients who discontinued treatment**	**4**		**7**		**5**		**3**	
Treatment discontinuation due to toxicity	4		2		2			
Treatment discontinuation due to progression			3		1		3	
Treatment discontinuation due to consent withdrawal			1		2			
Treatment discontinuation due to concurrent illness			1					

AE: adverse event; AESI: adverse event of special interest; DLT: dose-limiting toxicity; R-DHAOx: rituximab, dexamethasone, cytarabine, and oxaliplatin; R-DHAP: rituximab, dexamethasone, cytarabine, and cisplatin; SAE: serious adverse event.

**Table 3 cancers-14-01761-t003:** DLTs occurring during Cycle 1 (DLT set).

R-I-DHAP		R-I-DHAOx	
**Group A: Concomitant Administration: 13 Patients**		**Group B: Concomitant Administration: 12 Patients**	
Ibrutinib 420 mg/day: 6 Patients		Ibrutinib 420 mg/day: 6 Patients	
Patient 1	Grade 4 cutaneous eruption lasting 5 days	Patient 1	Grade 3 prostate infection lasting 11 days
Patient 2	Grade 4 sepsis lasting 10 days		Grade 3 cutaneous eruption lasting 10 days
Patient 3	Grade 4 thrombocytopenia lasting 7 days ^£^	Patient 2	Grade 3 cutaneous infection lasting 10 days
			Grade 3 febrile neutropenia lasting 1 day
		Patient 3	Grade 4 thrombocytopenia lasting 14 days
Ibrutinib 280 mg/day: 7 patients		Ibrutinib 280 mg/day: 6 patients	
Patient 4	Grade 4 neutropenia lasting 12 days	Patient 4	Grade 3 epigastric pain lasting 11 days
Patient 5	Grade 4 gastric hemorrhage lasting 13 days		
	Grade 4 thrombocytopenia lasting 13 days		
	Grade 3 atrial fibrillation lasting 2 days		
**Group A’: Intermittent administration: 10 patients ***		**Group B’: Intermittent administration: 12 patients**	
Ibrutinib 420 mg/day: 4 patients		Ibrutinib 420 mg/day: 6 patients	
	No DLT	Patient 1	Grade 3 hepatitis lasting 141 days
Ibrutinib 560 mg/day: 6 patients		Ibrutinib 560 mg/day: 6 patients	
Patient 1	Grade 3 acute renal insufficiency lasting 2 days		No DLT
Patient 2	Grade 3 hyponatremia lasting 1 day		
	Grade 3 nausea lasting 19 days		
	Grade 3 hypophosphatemia lasting 1 day.		
Patient 3	Grade 3 acute renal insufficiency lasting 5 days		
	Grade 3 vomiting lasting 17 days		

DLT: dose-limiting toxicity; R-DHAOx: rituximab, dexamethasone, cytarabine, and oxaliplatin; R-DHAP: rituximab, dexamethasone, cytarabine, and cisplatin. £: All DLTs recovered without sequelae, except this one, which was ongoing until death. *: Three patients received less than 80% of the planned dose of ibrutinib during Cycle 1 (renal insufficiency: 2; thyroid cancer: 1).

## Data Availability

The data presented in this study are available on request from the first author (C.B.).

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
