# Peer review of "Ibrutinib Associated with Rituximab-Platinum Salt-Based Immunochemotherapy in B-Cell Lymphomas: Results of a Phase 1b-II Study of the LYSA Group"

_cancers, 2022, doi:10.3390/cancers14071761_

Round 1

Reviewer 1 Report

The authors report the results of a Phase 1b-II Study of the LYSA Group and state that despite a strong rationale for combining ibrutinib and R-DHAP/R-DHAOx as both target lymphoma B-cells by different mechanisms, this approach was limited by significant toxicities.

Point to be addressed:

I would suggest slightly restructuring the rhetoric underlying the manuscript taking into account:

-Population, 

-Intervention

  • Comparison
  • Outcome
  • Time/Treatment2. how would the authors comment of the survival curve? This is something disappointing from a clinical perspectives and warrants an alternative trial design: the underlying message here is that a more personalized and tailored approach would be useful
  • 3. In the frame of this thinking, the authors mentioned and referenced (reff. 9, 18, 19, 22)the CD20-driven different approaches, nonetheless, this reviewer personally misses some important elements in the introduction/discussion section: the emerging use of bispecific antibodies, especially those of the Bispecific T cell Engager (BiTE) format, which engages B and T cells at the same timerelies on a sophisticated antibody design which, by augmenting the cytostatic effect, offers new intriguing opportunities. Key features of BiTE antibodies include the ability to redirect target cell lysis via T-cells at sub-picomolar concentrations, to activate T-lymphocyte killing in the presence of target cells and to allow T-cells to lyse target cells in series. This sequence of actions points to their capacity to elicit activation of T-cells that increase the number of CD4+ and CD8+ cytotoxic cells as long as target cells are accessible. Bi-specificity for CD3, for which stimulation is restrictive, intensifies the T-cell signal (please refer to PMID: 26818572 and expand).

Author Response

Dear Reviewer,

Many thanks for all your comments. The quality of our manuscript is now well improved.

Please find below our responses. 

  1. I would suggest slightly restructuring the rhetoric underlying the manuscript taking into account:

-Population, -Intervention, -Comparison, -Outcome, -Time/Treatment

While this would be a potentially interesting approach, we would like to stick to the usual format reporting phase 1-2 trials in the literature.

  1. How would the authors comment of the survival curve? This is something disappointing from a clinical perspectives and warrants an alternative trial design: the underlying message here is that a more personalized and tailored approach would be useful.7

The survival rate observed was not markedly different from that observed in previous studies (Coral, NCIC-CTG12 and ORCHHARD), but did not appear really improved, hence margin for improvement still exist.

  1. In the frame of this thinking, the authors mentioned and referenced (reff. 9, 18, 19, 22)the CD20-driven different approaches, nonetheless, this reviewer personally misses some important elements in the introduction/discussion section: the emerging use of bispecific antibodies, especially those of the Bispecific T cell Engager (BiTE) format, which engages B and T cells at the same timerelies on a sophisticated antibody design which, by augmenting the cytostatic effect, offers new intriguing opportunities. Key features of BiTE antibodies include the ability to redirect target cell lysis via T-cells at sub-picomolar concentrations, to activate T-lymphocyte killing in the presence of target cells and to allow T-cells to lyse target cells in series. This sequence of actions points to their capacity to elicit activation of T-cells that increase the number of CD4+ and CD8+ cytotoxic cells as long as target cells are accessible. Bi-specificity for CD3, for which stimulation is restrictive, intensifies the T-cell signal (please refer to PMID: 26818572 and expand).

Thank you for this suggestion. We added a sentence in the introduction and in the conclusion regarding this emerging therapeutic option.

Reviewer 2 Report

  1. The authors state on lines 51-52 page 2 (or lines 67-68, in the abstract) "Despite a strong rational for combining ibrutinib with either R-DHAP or R-DHAOx ...". To enhance this topic (by using pharmacokinetic and/or pharmacodinamic data of the scientific literature), the authors should add in Introduction section few sentences on a possible sinergistic effect (regarding anti-lymphomatous therapeutic efficacy) between ibrutinib and Rituximab-Platinum-Salt-containing regimens.
  2. In the Materials and Methods section, the authors should add data regarding Ethical Committee approval, informed consent from the included patients, and Helsinki Declaration.
  3. Please, improve English language.
  4. In Materials and Methods section and in the Results section, the authors should better specify the supportive treatment given in the cohorts of patients, in particular: prymary prophylaxis with G-CSF or long-acting G-CSF, and/or antibiotics, antifungal drugs and/or antiviral drugs.
  5. In Materials and Methods section, the authors state "Exploraty endpoint comprased feasibility of stem cell collection ..." line 189-page 4. Thus, in the Results section, the authors should report how Rituximab-Platinum-Salt-containing regimen impacted on peripheral blood stem cell collection. 

Author Response

Dear Reviewer,

Many thanks for all your comments. The quality of our manuscript is now well improved.

Please find below our responses. 

  1. The authors state on lines 51-52 page 2 (or lines 67-68, in the abstract) "Despite a strong rational for combining ibrutinib with either R-DHAP or R-DHAOx ...". To enhance this topic (by using pharmacokinetic and/or pharmacodinamic data of the scientific literature), the authors should add in Introduction section few sentences on a possible sinergistic effect (regarding anti-lymphomatous therapeutic efficacy) between ibrutinib and Rituximab-Platinum-Salt-containing regimens.

Thank you for this suggestion. We have now summarized the data of the literature regarding ibrutinib and others drug in the introduction: Data in ovarian cancer suggest synergic effects for cisplatin and ibrutinib. In CLL, Waldenström’s Macroglobulinemia and MCL, combination of ibrutinib and rituximab and bendamustinbe was associated with improvement of response and survival rates. Overall, those constitute a strong enough rationale to envision enhancing the quality of treatment responses before transplantation in relapsed/refractory non-Hodgkin B-cell lymphoma (R/R B-NHL) patients, by conducting a Phase 1b-II study to further evaluate ibrutinib given in combination with either R-DHAP or R-DHAOx.”

  1. In the Materials and Methods section, the authors should add data regarding Ethical Committee approval, informed consent from the included patients, and Helsinki Declaration.

We added the information in the section “2.2 Procedures”

The study (NCT02055924) was sponsored by LYSARC (the Lymphoma Study Clinical Organisation) and conducted according to the Declaration of Helsinski and the International Conference on Harmonization Guidelines for Good Clinical Practice. The protocol was approved by the Comité de Protection des Personnes (Ethics Committee) Sud-Est II, Lyon, in France and the Comité d’Ethique in Belgium. All patients provided a written informed consent.

  1. Please, improve English language.

The draft was reviewed by a professional English Editor service before submission.

  1. In Materials and Methods section and in the Results section, the authors should better specify the supportive treatment given in the cohorts of patients, in particular: primary prophylaxis with G-CSF or long-acting G-CSF, and/or antibiotics, antifungal drugs and/or antiviral drugs.

We added these precisions in the section “2.2 Procedures”

All patients received mandatory primary infectious prophylaxis by granulocyte grow factor injections (filgrastim or pegfilgrastim) and must receive prophylactic antibiotics (sulfamethoxazole/trimethoprim) and antiviral therapy (valaciclovir) according to local practice at each center.”

  1. In Materials and Methods section, the authors state "Exploratory endpoint comprased feasibility of stem cell collection ..." line 189-page 4. Thus, in the Results section, the authors should report how Rituximab-Platinum-Salt-containing regimen impacted on peripheral blood stem cell collection. 

Many thanks for this remark. We added the following sentence in the section “3.2 Patients outcome”

One patient out of the 35 mobilized, 4 patients did not reach sufficient progenitor cells collection yield to proceed to ASCT. A median of 5.29 106 (0.90; 35.72) CD34+ cells were obtained in the 31 collected patients after one (24 patients), two (5 patients) or three (2 patients) apheresis.”

Round 2

Reviewer 1 Report

The authors have clarified several of the questions I raised in my previous review. Most of the major problems have been addressed by this revision.